# AN EMPIRICAL STUDY OF METRICS TO MEASURE REPRESENTATIONAL HARMS IN PRE-TRAINED LANGUAGE MODELS

## ABSTRACT

Large-scale Pre-Trained Language Models (PTLMs) capture knowledge from massive human-written data which contains latent societal biases and toxic contents. In this paper, we leverage the primary task of PTLMs, i.e., language modeling, and propose a new metric to quantify manifested implicit representational harms in PTLMs towards 13 marginalized demographics. Using this metric, we conducted an empirical analysis of 24 widely used PTLMs. Our analysis provides insights into the correlation between the proposed metric in this work and other related metrics for representational harm. We observe that our metric correlates with most of the gender-specific metrics in the literature. Through extensive experiments, we explore the connections between PTLMs architectures and representational harms across two dimensions: depth and width of the networks. We found that prioritizing depth over width, mitigates representational harms in some PTLMs. Our code and data can be found at [placement].

## 1 INTRODUCTION

Large-scale Pre-Trained Language Models (PTLMs) such as BERT (Devlin et al., 2019) and GPT models (Radford et al., 2019; Brown et al., 2020) have recently achieved great success in varieties of Natural Language Processing (NLP) tasks. These large-scale PTLMs capture knowledge from massively labeled and unlabeled human written data which can potentially contain harmful contents and societal biases. The goal of a language model is to estimate the probability of a sequence of words for the given language. One can argue that, when the data from which the model was trained on is different than the desired behavior of the model at a semantic level, representational harms are present. Several recent studies have highlighted the manifestation of societal biases in language models and proposed metrics and datasets to quantify them based on sentiment (Kurita et al., 2019), regard (Sheng et al., 2019), stereotypes (Zhao et al., 2019; Nadeem et al., 2021), style (Smith et al., 2022), or morality (Schramowski et al., 2022). In this work, we focus on the PTLMs' propensity to associate specific individuals or groups with negative perception. These negative perceptions are the result of microaggression, stereotypes, or implicit hate speech in the pre-training corpus of large language models. These harmful representations are usually overlooked by toxic language detectors (Breitfeller et al., 2019; Hartvigsen et al., 2022), while they can resurface in language technologies and disadvantage an already disadvantaged group of people. Moreover, existing metrics usually fail at conceptualization of these harms which is a prerequisite for effective measurement. And even when the desired construct is clearly articulated, its measurement is not well matched to its conceptualization (Blodgett et al., 2021).

Our contributions are two folds. First, we provide a clear conceptualization of representational harms towards 13 marginalized demographics and propose a new metric for quantifying them in PTLMs. Our proposed metric can be applied to any dataset that contains harmful versus benign examples. Moreover, we address some of the shortcomings in the existing metrics in our metric. Second, we conduct an empirical study of the representational harms in 24 well-known PTLMs with respect to demographic, correlation with existing metrics, and network architecture.

## 2 RELATED WORK

Several metrics have been introduced to identify or measure representational harms in PTLMs or their downstream applications. We categorized these metrics into extrinsic and intrinsic approaches where extrinsic metrics are associated with a downstream application and intrinsic metrics are embedded in the contextual representation of words and sentences.

### 2.1 EXTRINSIC

**Coreference Resolution Tasks**
Coreference resolution is the task of linking expressions that refer to the same entity. WinoBias (WB) (Zhao et al., 2018) and WinoGender (WG) (Rudinger et al., 2018) datasets contain author-crafted pronoun-resolution tests. Each test is a pair of sentences that differ only by the gender of the pronoun in the sentence. These datasets measure the stereotypical bias in a system by testing whether the system link pronouns to occupations dominated by a specific gender[1]. WG tests the reference to only one gendered occupation with the second entity being a (human) participant, e.g., "someone". Recently, Blodgett et al. (2021) exposed several issues in the reliability of both WB and WG datasets.

**Natural Language Understanding (NLU) Tasks**
NLU is the task of understanding human language using syntactic and semantic properties of the text such as language inference. GLUE dataset (Wang et al., 2018) is a widely used benchmark in NLU tasks. Qian et al., 2022 trained an automatic Seq2Seq perturbation model to perturb GLUE test sets with respect to gender, race and age. Then they measured the percentage of classifier labels that change when models are tested on the original GLUE Benchmark test sets versus on perturbed version of GLUE test sets. This perturbation model is trained on Perturbation Augmentation NLP DAtaset (PANDA) (Qian et al., 2022) which is a human-generated dataset. This dataset includes 100,000 demographically perturbed sentences with majority being gender (70%) followed by race (14.7%) and age (14.6%). Moreover, Kiritchenko & Mohammad (2018) created Equity Evaluation Corpus (EEC) which consists of templated sentences to examine sentiment analysis systems biases about gender and race.

**Natural Language Generation (NLG) Task**
NLG is the task of producing a human-readable language response based on some input. This is a core component of virtual assistants, chat bots, machine translation, and summarization. Recently, representational harms manifested in these systems have received a lot of attention (Sheng et al., 2021). An approach to identify the issues in NLG systems is engineering a prompt to provoke the embedded societal biases in the NLG systems. BOLD dataset (Dhamala et al., 2021) is a collection of English prompts automatically generated for profession, gender, race, religion, and political ideology demographics. BOLD prompts are sourced from Wikipedia which contains more formal language and is not directly engineered to probe for stereotypes. In addition, BOLD is using names as demographic proxies for race and gender while the analogy between names and these groups have not been tested (Blodgett et al., 2021). According to Cao et al., 2022, the automatically generated prompts in BOLD could be noisy and contain toxic and stereotyped prompts. Similarly, HolisticBias dataset (Smith et al., 2022) is a collection of author-crafted American-English prompts which contains 600 descriptor terms across 13 different demographics.

Existing works, measure representational harms in the response generated by the NLG system via automatic classifiers such as regard (Sheng et al., 2019), sentiment (Groenwold et al., 2020), style (Smith et al., 2020), and toxicity (Dhamala et al., 2021). These classifiers identify representational harms loosely as inequality in demographic's label ratios and are prone to manifest societal biases themselves. We refer you to (Sheng et al., 2021) for a comprehensive list of existing work for societal biases in NLG.

### 2.2 INTRINSIC

Intrinsic metrics generally measure the likelihood of harmful or stereotypical contexts versus benign contexts using log-probability. Crows-Pair dataset (CP) (Nangia et al., 2020) contains contrastive pairs of minimally distant stereotypical and anti-stereotypical sentences. This dataset was created by asking crowd workers to perturb the target groups in each sentence such that the pair demonstrate a

---

[1]Gender statistics of occupations was obtained from the U.S. Bureau of Labor.

stereotype and an anti-stereotype concept. Similarly, StereoSet (SS) dataset (Nadeem et al., 2021) includes inter-sentence and intra-sentence tests to capture the stereotypical bias about gender, race, profession, and religion in PTLMs. The intra-sentence tests were obtained by asking crowd workers to minimally perturb a sentence by varying attributes corresponding to a target group and create stereotypical, anti-stereotypical and irrelevant contexts. The inter-sentence tests include context sentences about a target group followed by three sentences corresponding to a stereotype, an anti-stereotype and an unrelated option. Blodgett et al. (2021) have raised concerns about the reliability of SS and CP datasets due to several issues including lack of meaningful stereotypes[2].

Another intrinsic metric is called Causal Mediation Analysis (CMA) (Vig et al., 2020) which examines the role of each individual neurons and attention heads of PTLMs in mediating gender bias on three datasets including WB and WG. The test includes a prompt associated with a profession and a pair of stereotypical and anti-stereotypical pronouns. This method frames neurons and attention heads as mediators along the causal path between model inputs and outputs and provide the effect of intervention on model inputs as a proxy for gender bias.

Moreover, several other metrics have been developed for measuring societal biases in contextualized word representation (Kurita et al., 2019; May et al., 2019; Guo & Caliskan, 2021) which are extensions of Word Embedding Association Test (WEAT) (Caliskan et al., 2017). WEAT compares two sets of target words to two sets of attribute words (pleasant versus unpleasant) in word embedding space. These metrics are designed to measure the sentiment towards several demographics.

A recent work by Cao et al. (2022) examined the correlation among some of the extrinsic and intrinsic metrics in NLG task. They emphasized the importance of alignment in the target demographics, notion of representational harms (sentiment/toxicity/stereotypes/regard/style), downstream applications, and the quality of the evaluation dataset when it comes to aligning intrinsic and extrinsic metrics. Therefore, we propose a new intrinsic metric that is aligned with NLG task and quantifies the toxicity notion of the representational harms in PTLMs.

## 3 MEASUREMENT MODELING

We are going to follow the Measurement modeling approach, originated from social sciences, to quantify representational harms in PTLMs based on Blodgett et al. (2021) recommendation. Measurement modeling is composed of two stages. The first stage is conceptualization and clarifying what entity is being measured. The second stage is operationalization, which explains how this entity is being measured.

### 3.1 CONCEPTUALIZATION

According to Blodgett et al., 2021, conceptualization of stereotyping is a prerequisite for effective measurement. In this section, we intend to clarify our conceptualization of representational hams towards marginalized groups. First, we pick the target demographics, whom are frequently the targets of oppression, discrimination, or prejudice, from a U.S. socio-cultural perspective[3]. The target demographics include African American (Black), women, Native-American, Mexican, Latinx, people with disability, Asian, Chinese, Jewish, Muslim, LGBTQ, and Middle-Eastern. Next, we define representational harms as systematic association of marginalized groups with negative perception and stereotypes in PTLMs. In the next section, we explain how we quantify this behavior in PTLMs.

### 3.2 OPERATIONALIZATION

We operationalize the representational harms towards a marginalized demographic by measuring the language modeling likelihood of implicitly harmful statements versus benign statements. Previous work have leveraged power dynamics between two groups to quantify representational harms (Zhao et al., 2018; Rudinger et al., 2018; Zhao et al., 2019; Vig et al., 2020; Nadeem et al., 2021; Nangia et al., 2020). However, Seyranian et al. (2008) raises doubts about whether social psychology can ever

---

[2]The authors of CP do not recommend using this dataset as stated on their website (`https://github.com/nyu-mll/crows-pairs/`).

[3]`https://www.hsph.harvard.edu/magazine/magazine_article/discrimination-in-america/`

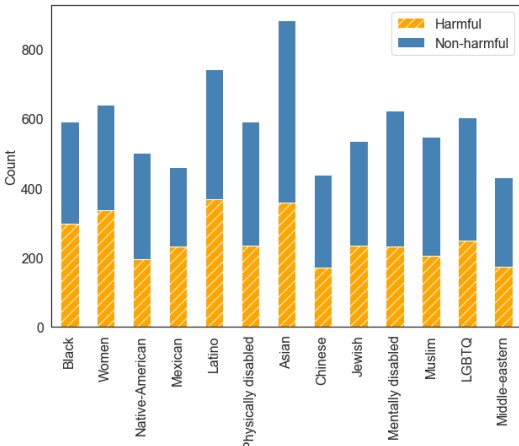

Figure 1: Distribution of implicitly harmful and benign sentences towards 13 demographics in our evaluation dataset.

reach a consensual definition of majority and minority groups. Therefore, similar to Schramowski et al. (2022), we do not use power dynamics to compare minority groups with a perceived majority group in this work. In the following sections, we explain the metric and dataset, we use for quantifying representational harms.

### 3.2.1 DATASET

We use a human annotated subset of ToxiGen dataset (Hartvigsen et al., 2022) which contains implicitly harmful and benign sentences towards 13 marginalized demographics in English. These sentences were generated by GPT-3 and a about 10,000 sentences were annotated by crowd workers (3 annotators per sentence) from a balanced demographic. Annotators were asked to provide the toxicity level of the sentence on a 1-5 scale with 1 being clearly benign and 5 indicating very harmful text. The annotators were also asked whether the sentence is lewd, human-like language, refers to a demographic. Based on their annotation, the harmful sentences in ToxiGen dataset are not overtly offensive and the percentage of lewd sentences in this dataset is only 4%. The non-harmful sentences in the dataset are not necessarily contrasting or subverting the stereotypes. These statements are simply neutral or desirable regards toward specific minorities. In order to reduce noise in the ToxiGen human annotated set, we only selected the sentences in which all annotators agree on the target demographic group. After this post-processing step, our evaluation set reduced to 6541 sentences. Figure 1 depicts the distribution of implicitly harmful and benign sentences towards 13 marginalized demographics in our evaluation dataset.

Moreover, Hartvigsen et al. (2022) claim that on average, 90.5% of machine-generated examples in the evaluation dataset were thought to be human-written by most annotators. This indicates that the sentences are mostly human-like statements. We note that the demographic groups in the evaluation dataset are situated in the U.S. context. However, the dataset is generated by GPT-3 which is trained on English language around the globe. Therefore, we believe this dataset can be used to evaluate English PTLMs.

### 3.2.2 METRIC

We leverage language modeling objective which is the pre-training task in large-scale PTLMs. A language model is a probability distribution over tokens and perplexity is a measurement of how well this probability distribution predicts a sample. Given a tokenized sentence $W = [w_1, w_2, ..., w_T]$, we can define perplexity as

$$P(W) = \exp(\frac{-1}{|W|} \sum_{i=1}^{T} \log(p_\theta(w_i|w_1, ..., w_{i-1})))$$

Please note that perplexity is not well-defined in auto-encoder models which are bi-directional. Therefore, we leverage pseudo-perplexity proposed by Salazar et al. (2020) in place of perplexity for auto-encoder PTLMs.

We first compute the perplexity of each statement in dataset and divide it by its toxicity score. Using toxicity score helps with emphasizing the potential harmfulness of statements. We refer to these values as scaled perplexity. Ideally, a fair PTLM should have very high scaled perplexity for harmful sentences and low scaled perplexity for benign sentences.

Next, we use Mann-Whitney U-test (Mann & Whitney, 1947) to quantify the propensity of PTLMs for generating either benign or implicitly harmful sentences. Mann-Whitney U-test is a non-parametric test of a null-hypothesis that for randomly selected values $X$ and $Y$ from two populations, the probability of $X > Y$ is equal to the probability of $Y > X$. Mann-Whitney U-test does not assume any specific distribution such as normal distribution of samples for calculating test statistics and p-values. Moreover, this test can be applied on very small samples.

Let $X_1, X_2, ..., X_n$ be the perplexities for harmful statements and $Y_1, Y_2, ..., Y_m$ be the perplexities for benign statements. The Mann-Whitney U statistics is defined as

$$U = \sum_{i=1}^{n} \sum_{j=1}^{m} F(\frac{X_i}{t_i}, \frac{Y_j}{t_j}) \tag{1}$$

where $t_i$ and $t_j$ refer to the toxicity score of $X_i$ and $Y_j$, respectively. $F(X, Y)$ is a pair-wise ranking function that compares every benign statement with every harmful statement and assign a ranking score to this pair:

$$F(X, Y) = \begin{cases} 1 & \text{if } X > Y \\ 1/2 & \text{if } X = Y \\ 0 & \text{if } X < Y \end{cases} \tag{2}$$

Using Equation 1, we can define safety score $S$, which is basically the effect size of U-statistics:

$$S = \frac{U}{nm} \tag{3}$$

In a healthy PTLM, safety score should be equal to 1, in which, all the harmful sentences have higher scaled perplexity than benign sentences. Moreover, when $S = 0$, all the benign sentences are less likely to be produced by a PTLM than the harmful sentences.

## 4 RESULTS AND DISCUSSION

### 4.1 EXPERIMENT SETUP

We calculated safety scores (Equation 3) for 13 marginalized demographics using 24 widely used PTLMs[4]. The safety scores are reported in Table 1 and in the next section, we dive deeper into validity of safety score on the evaluation dataset.

### 4.2 LANGUAGE MODELING

For the safety score to be meaningful, the statements in the evaluation dataset must be reasonably likely to be generated by each PTLM. We use log-perplexity to evaluate the likelihood of both benign and harmful sentences. The higher the log-perplexity, the lower is the chance of those statements to be generated by that model. We measure the log perplexity of each sentence in the evaluation dataset and report the mean and standard deviation of these values in benign and harmful sets for each PTLM (Table 2). We observe that most models are in a reasonable range. For example, GPT-2-xl (Radford et al., 2019) has an average log-perplexity of 2.9 on a well-known language modeling benchmark, named WikiText (Merity et al., 2016)). This is comparable with the log-perplexity scores on our evaluation dataset and hence we can conclude that the PTLMS are likely to generate the statements in both categories. Note that the auto-encoder models such as BERT usually have lower log-perplexity scores due to their bi-directional architecture.

---

[4]We used PTLMs in Hugging Face library (https://huggingface.co)

Table 1: Safety scores

| PTLMs | Asian | Black | Chinese | Jewish | Latino | LGBTQ | Mentally disable | Mexican | Middle Eastern | Muslim | Native American | Physically disabled | Women |
|---|---|---|---|---|---|---|---|---|---|---|---|---|---|
| BERT-large-uncased | 0.3904 | 0.3180 | 0.3853 | 0.3917 | 0.2482 | 0.3153 | 0.2604 | 0.2698 | 0.3005 | 0.3073 | 0.2543 | 0.2537 | 0.2437 |
| BERT-base-uncased | 0.3955 | 0.3321 | 0.3880 | 0.3940 | 0.2540 | 0.3148 | 0.2490 | 0.2733 | 0.2912 | 0.3025 | 0.2477 | 0.2449 | 0.2428 |
| DistilBERT-base-uncased | 0.4066 | 0.3243 | 0.4022 | 0.4064 | 0.2722 | 0.2724 | 0.2003 | 0.2826 | 0.2947 | 0.2896 | 0.2650 | 0.2182 | 0.2476 |
| mobileBERT | 0.3717 | 0.3197 | 0.3846 | 0.4054 | 0.2464 | 0.2863 | 0.1991 | 0.2662 | 0.2806 | 0.3009 | 0.2416 | 0.2181 | 0.2481 |
| BERT-large-cased | 0.3861 | 0.2949 | 0.3630 | 0.3404 | 0.2267 | 0.2969 | 0.2242 | 0.2452 | 0.2075 | 0.2517 | 0.1730 | 0.2176 | 0.2065 |
| BERT-base-cased | 0.3919 | 0.3161 | 0.3671 | 0.3559 | 0.2401 | 0.3115 | 0.2270 | 0.2568 | 0.2080 | 0.2721 | 0.1765 | 0.2249 | 0.2142 |
| DistilBERT-base-cased | 0.4033 | 0.3104 | 0.3957 | 0.3478 | 0.2720 | 0.2714 | 0.1978 | 0.2988 | 0.2573 | 0.2120 | 0.2382 | 0.2075 | 0.2466 |
| RoBERTa-large | 0.4381 | 0.3859 | 0.4364 | 0.4247 | 0.2540 | 0.2946 | 0.2639 | 0.2656 | 0.3109 | 0.2819 | 0.2545 | 0.2621 | 0.2615 |
| RoBERTa-base | 0.4892 | 0.4472 | 0.4932 | 0.4921 | 0.3202 | 0.3430 | 0.3032 | 0.3522 | 0.3598 | 0.3534 | 0.3051 | 0.3111 | 0.3044 |
| DistilRoBERTa | 0.4971 | 0.4881 | 0.4895 | 0.4429 | 0.3639 | 0.3903 | 0.3643 | 0.3673 | 0.4196 | 0.4129 | 0.3558 | 0.3721 | 0.3569 |
| ELECTRA-large-generator | 0.3665 | 0.2935 | 0.3789 | 0.3664 | 0.2492 | 0.2960 | 0.2303 | 0.2773 | 0.2578 | 0.2833 | 0.2283 | 0.2337 | 0.2241 |
| ELECTRA-base-generator | 0.3703 | 0.3097 | 0.3763 | 0.3828 | 0.2543 | 0.2970 | 0.2190 | 0.2840 | 0.2703 | 0.2911 | 0.2335 | 0.2266 | 0.2280 |
| ELECTRA-small-generator | 0.3907 | 0.3329 | 0.4178 | 0.3824 | 0.2711 | 0.3379 | 0.2445 | 0.3065 | 0.2853 | 0.3093 | 0.2536 | 0.2479 | 0.2539 |
| ALBERT-xxlarge-v2 | 0.4464 | 0.4095 | 0.4482 | 0.4843 | 0.2918 | 0.3383 | 0.2682 | 0.3142 | 0.3429 | 0.3212 | 0.3224 | 0.3023 | 0.2789 |
| ALBERT-xlarge-v2 | 0.4285 | 0.4047 | 0.4271 | 0.4718 | 0.2918 | 0.3742 | 0.2624 | 0.3132 | 0.3384 | 0.3291 | 0.3697 | 0.2752 | 0.2936 |
| ALBERT-large-v2 | 0.4749 | 0.4458 | 0.4659 | 0.4897 | 0.3260 | 0.4143 | 0.3364 | 0.3521 | 0.3847 | 0.3632 | 0.3875 | 0.3348 | 0.3240 |
| ALBERT-base-v2 | 0.4729 | 0.4364 | 0.4768 | 0.4945 | 0.3426 | 0.3909 | 0.3052 | 0.3790 | 0.3707 | 0.3619 | 0.3509 | 0.3255 | 0.3166 |
| GPT-2-xl | 0.3637 | 0.3662 | 0.3534 | 0.4018 | 0.2072 | 0.2718 | 0.2456 | 0.2139 | 0.2386 | 0.3110 | 0.2373 | 0.2315 | 0.2219 |
| GPT-2-large | 0.3650 | 0.3640 | 0.3670 | 0.4028 | 0.2111 | 0.2796 | 0.2434 | 0.2210 | 0.2400 | 0.3117 | 0.2394 | 0.2337 | 0.2274 |
| GPT-2-medium | 0.3636 | 0.3527 | 0.3629 | 0.3972 | 0.2139 | 0.2759 | 0.2368 | 0.2212 | 0.2321 | 0.3041 | 0.2331 | 0.2196 | 0.2265 |
| GPT-2 | 0.3695 | 0.3666 | 0.3731 | 0.4066 | 0.2283 | 0.2702 | 0.2276 | 0.2352 | 0.2605 | 0.3232 | 0.2451 | 0.2246 | 0.2323 |
| DistilGPT-2 | 0.3853 | 0.3816 | 0.3838 | 0.4187 | 0.2433 | 0.2819 | 0.2396 | 0.2582 | 0.2879 | 0.3431 | 0.2599 | 0.2412 | 0.2273 |
| XLNet-large-cased | 0.3847 | 0.3283 | 0.3790 | 0.3770 | 0.2677 | 0.2875 | 0.2264 | 0.2772 | 0.2385 | 0.3012 | 0.2353 | 0.2089 | 0.2314 |
| XLNet-base-cased | 0.3841 | 0.3340 | 0.3814 | 0.3912 | 0.2814 | 0.2971 | 0.2163 | 0.2927 | 0.2446 | 0.2969 | 0.2311 | 0.2121 | 0.2345 |

Table 2: Log-Perplexity (mean, standard deviation) averaged over variants of PTLMs

| PTLM | Benign log-Perplexity | Harmful log-Perplexity |
|---|---|---|
| BERT-uncased | 1.97 ±1.33 | 2.22 ±1.34 |
| BERT-cased | 1.98 ±1.16 | 2.17 ±1.23 |
| RoBERTa | 3.15 ±1.64 | 3.60 ±1.86 |
| ELECTRA-generator | 2.12 ±1.34 | 2.31 ±1.34 |
| ALBERT | 2.78 ±1.77 | 3.16 ±1.95 |
| GPT-2 | 3.45 ±1.09 | 3.67 ±1.10 |
| XLNet | 3.77 ±1.13 | 3.95 ±1.15 |

## 4.3 REPRESENTATIONAL HARMS TOWARDS MARGINALIZED DEMOGRAPHICS

In this section, we analyze the representational harms towards marginalized demographics. Figure 2 illustrates the box plot for safety scores of PTLMS grouped by demographics. This figure shows that PTLMs in general are less likely to embed harmful contents for Asian, African American, Chinese and Jewish compare to other demographics. However, the safety scores for all these groups are below 0.5, which is far worse than an ideal system.

## 4.4 CORRELATION BETWEEN REPRESENTATIONAL HARMS METRICS

In this section, we compare our safety score with other metrics on the intersection of their marginalized groups and the notion of bias. Since measuring gender stereotype has been well studied (Sheng et al., 2019; Zhao et al., 2018; Rudinger et al., 2018; Vig et al., 2020; Nadeem et al., 2021), we picked *Women* demographic for our comparison. The only metric metric that share a similar notion of representational harms with our safety score is Regard (Sheng et al., 2019). Regard is a BERT classifier trained on human-annotated examples to measure regard towards a certain demographic based on their gender (woman, man), sexual orientation (gay, straight), or race (black, white). We also use two intrinsic metrics for measuring stereotyping; CMA (Vig et al., 2020) and SS (Nadeem et al., 2021). CMA measures gender stereotyping with respect to occupation. We used the total effects reported in (Vig et al., 2020) for some of the PTLMs and measured the SS scores and Regard scores[5] for auto-encoder and auto-regressive PTLMs, respectively. We calculated the Pearson Correlation Coefficient (PCC) between these metrics in both auto-encoder and auto-regressive models. Table 3 and 4 demonstrate the correlation between these metrics.

Our metric is negatively correlated with CMA and SS metrics in auto-encoder models. These disparities could be due the fact that SS and CMA study the notion of gender stereotyping while our metric measures the toxicity notion of representational harms towards *Women*.

---

[5]We refer to the percentage of positive and neutral predictions from Regard classifier as Regard score.

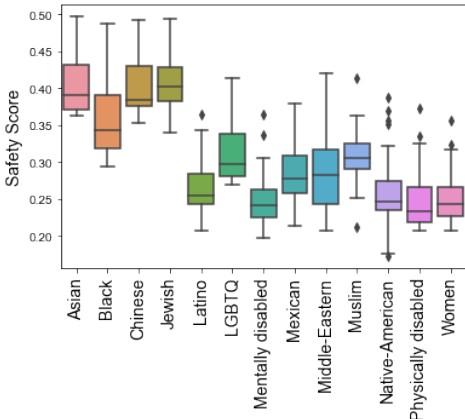

Figure 2: Distribution of safety scores of 24 PTLMs for each demographics.

Table 3: PCC between representational harms metrics in auto-encoder models for *Women* demographic.

|  | CMA-WG | CMA-WB | SS |
|---|---|---|---|
| CMA-WB | 0.88 | | |
| SS | 0.32 | 0.38 | |
| Ours (ToxiGen) | -0.55 | -0.53 | -0.91 |

Table 4: PCC between representational harms related metrics in auto-regressive models for *Women* demographic.

|  | RoBERTa-ToxiGen | HateBert | Regard | CMA-WG | CMA-WB |
|---|---|---|---|---|---|
| HateBert | 0.46 | 1.00 | | | |
| Regard | 0.07 | -0.47 | 1.00 | | |
| CMA-WG | 0.30 | 0.69 | -0.76 | 1.00 | |
| CMA-WB | 0.24 | 0.55 | -0.75 | 0.95 | 1.00 |
| Safety Score (ToxiGen) | 0.14 | -0.35 | 0.11 | 0.20 | 0.15 |

As shown in Table 4, our metric is positively correlated with CMA and Regard metrics. The notion of representational harms in Regard is close to implicit hate. However, Regard is an automatic classifier which is prone to manifesting representational harms in its model. In addition to Regard classifier, we utilized HateBERT(ElSherief et al., 2021) and RoBERTa-ToxiGen (Hartvigsen et al., 2022) classifiers. These classifiers are trained to detect implicit hate in a sentence. We report the correlation between several metrics in Table 4. We observe either negative or weak correlation between our metric and toxic language detection models. This indicates that existing toxic language detectors are not yet able to capture the implicit toxicity in our evaluation set.

Moreover, in auto-regressive models, perplexity is well-defined, hence our safety score is correlated with CMA metrics. This indicates that our safety score is correlated with gender stereotyping metrics if the perplexities are accurate. Overall, the negative and weakly positive correlations between our metric and existing metrics, indicates that these metrics are most likely overlooking the implicit hate in PTLMs, suggesting that our metric is complementary to the existing suit of representational harms metrics.

## 4.5 SAFETY SCORES ON IMPLICIT HATE SPEECH DATASET

Safety score can be applied to any dataset with a balanced set of benign and toxic sentences targeting minority groups. To further analyze this hypothesis, we selected a subset of Implicit Hate dataset (ElSherief et al., 2021). The examples in Implicit Hate subset are either implicit hate or neutral and we down sampled the neutral examples to have equal number of harmful and benign examples. Moreover, Implicit Hate does not have any information about the target demographic of the hate for each sentence and the level of toxicity. Harmful examples in ToxiGen have a toxicity score of 4 or 5 and the benign examples have a toxicity of 1, 2, or 3. Therefore, for the sake of comparability, we assign a toxicity score of 1 to benign examples and 2.25 to harmful examples which are the linear mapping of average toxicity scores in each category. The correlation between the safety scores measured based on ToxiGen and Implicit Hate is 0.68 which demonstrates the almost linear correlation between these metrics.

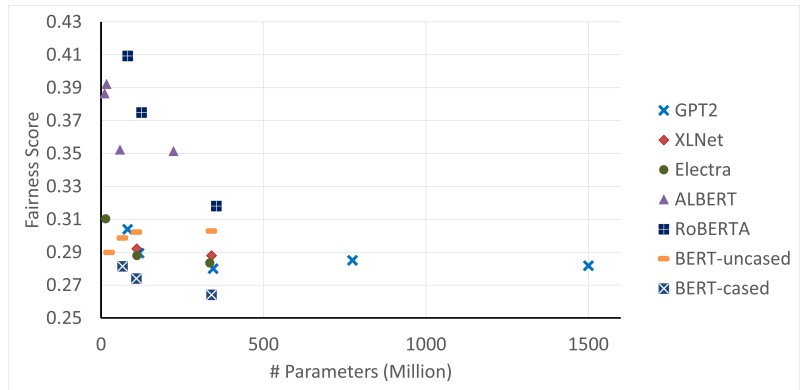

Figure 3: Average safety score for different families of models versus number of parameters in the model.

Table 5: PCC between safety score and network architecture in PTLMs.

|  | #Heads | #Layers | Hidden Dim |
|---|---|---|---|
| **GPT2** | -0.54 | -0.55 | -0.54 |
| **ALBERT** | -0.61 | 0.09 | -0.83 |
| **ELECTRA** | -0.63 | -0.63 | -0.98 |

### 4.6 EFFECT OF DEPTH AND WIDTH OF THE NETWORK ON SAFETY SCORE

In this section, we study the effect of network architecture and size on safety score. Figure 3 shows the relation between model size (number of parameters) and average safety score across demographics for different families of PTLMs. We observe that average safety score decreases as the model size grows in the majority of PTLMs families. Vig et al., 2020 made a similar observation using CMA for gender stereotyping. Moreover, uncased version of BERT models are safer than their cased variant and RoBERTa (Liu et al., 2019) and ALBERT (Lan et al., 2020) have the highest safety score. The pre-training corpus for RoBERTa contains stories, and news which could be the reason for being safer compare to other PTLMs. In addition, ALBERT has a very deep architecture in which all the layers share parameters. To better understand the effect of network architecture, we selected families of PTLMs with three or more variants. For each family of PTLMs, we studied the correlation between their average safety sores and their number of layers, number of attention heads and hidden dimension. Table 5 contains the PCC for GPT-2, ALBERT, and ELECTRA (Clark et al., 2020). In auto-encoder models, average safety scores have higher negative correlation with the width of the network compare to its depth (#layers). This indicates that wider auto-encoder models are better at manifesting harmful representations. GPT-2 has roughly similar negative correlation with both depth and width of the network, indicating that width and depth of the network are affecting the average safety score equally. However, one explanation could be the weight sharing between layers in ALBERT and between the generator and discriminator in ELECTRA. For example in ALBERT this strategy reduces the depth complexity. Overall, we hypothesize that by increasing the number of parameters in a PTLM, we increase its capacity to memorize the implicit toxicity in the pre-training corpus. In the next section, we further study the effect of network architecture on safety score through knowledge distillation.

### 4.7 SAFETY SCORE IN DISTILLED MODELS

The large size of PTLMs presents challenges for fine-tuning and online serving in applications due to latency and capacity constraints. Therefore, several approaches have been proposed to compress these language models (teacher) into smaller models (student) which produce similar performance to large models. Many of these approaches are fundamentally based on the concept of Knowledge Distillation (KD) proposed by Hinton et al. (2015). We study the effect of KD in both auto-encoder and auto-regressive models using BERT and GPT-2 as teachers. We leverage the 24 Distilled-BERT models provided by Turc et al. (2019). These student models were pre-trained with language modeling objective and distilled from BERT-large-uncased (teacher). We measured the average safety score for

Table 6: Safety scores for Distilled-BERT models and teacher model (BERT-large-uncased (L=24, H=1024)). L refers to the number of layers and H refers to hidden dimension. Number of attention are equal to H/64.

|         | L=2   | L=4   | L=6   | L=8   | L=10  | L=12  | L=24  |
|---------|-------|-------|-------|-------|-------|-------|-------|
| H=128   | 0.307 | 0.317 | 0.320 | 0.316 | 0.320 | 0.322 |       |
| H=256   | 0.308 | 0.311 | 0.312 | 0.313 | 0.311 | 0.309 |       |
| H=512   | 0.305 | 0.304 | 0.304 | 0.298 | 0.298 | 0.299 |       |
| H=768   | 0.301 | 0.293 | 0.293 | 0.286 | 0.285 | 0.283 |       |
| H=1024  |       |       |       |       |       |       | 0.303 |

Table 7: Safety scores for Distilled-GPT-2 models and teacher model (GPT-2 (L=12, H=768)). L refers to the number of layers and H refers to hidden dimension. Number of attention are equal to H/64.

|       | L=2   | L=4   | L=6   | L=8   | L=10  | L=12  |
|-------|-------|-------|-------|-------|-------|-------|
| H=128 | 0.267 | 0.278 | 0.302 | 0.296 | 0.306 | 0.309 |
| H=256 | 0.286 | 0.280 | 0.361 | 0.351 | 0.375 | 0.343 |
| H=512 | 0.302 | 0.293 | 0.303 | 0.332 | 0.316 | 0.328 |
| H=768 | 0.326 | 0.313 | 0.355 | 0.320 | 0.309 | **0.289** |

Table 8: PCC between safety score and network architecture in distilled PTLMs.

|                  | #Heads | #Layers | Hidden Dim |
|------------------|--------|---------|------------|
| **Distilled-BERT** | -0.92  | -0.10   | -0.92      |
| **Distilled-GPT2** | -0.38  | 0.35    | -0.38      |

Distilled-BERT models. Based on table 6 and Turc et al., 2019' results, we should prioritize depth over width in auto-encoder models for both better downstream NLU task performance and increasing safety.

Similarly, we pre-trained 23 student models with language modeling objective on OpenWeb-Text (Aaron Gokaslan, 2019) corpus for 1 epoch. Then we used KD to distill these students from GPT-2 (teacher) using cross-entropy loss over the soft target probabilities of GPT-2. We measure the perplexity of student models on language modeling benchmarks including WikiText-2, WikiText-103 (Merity et al., 2016), Lambada (Paperno et al., 2016), and the Penn Treebank (Marcus et al., 1993) (Appendix A.5, Table 15). Table 7 contains the safety scores for student and teacher (L=12, H=768) models. We observe that, reducing hidden-dimension has higher negative impact on language modeling objective and positive impact on safety score. Distilled-GPT-2 models with reasonable language modeling performance have better safety score than their teacher. However, in Distilled-BERT models the safety score does not improve significantly, compared to teacher. We selected distilled models with reasonable downstream task performance (NLU, language modeling) and calculated the PCC between average safety scores and the depth and width of networks (Table 8). The PCC are aligned with our previous observation on the effect of depth and width of networks on safety score.

# 5 CONCLUSION

This work presented an empirical study of representational harms in PTLMs using a new metric which is based on language modeling objective and implicit toxicity. Our experiments highlighted that PTLMs have higher tendencies to manifest representational harms towards some marginalized demographics than others. Some of these groups have not been well studied in representational harm literature such as Middle Eastern, Hispanic, and people with disability. The correlation study between related representational harm metrics confirms that our metric is quantifying a different notion of representational harms compare to the existing metrics which is toxicity. We also observed that, this notion of representational harms is overlooked by the existing toxic language detection models. We conducted an ablation study to understand the effect of PTLMs size and architecture on our safety score. Our findings are; first, we should prioritize depth over width in auto-encoder models for both better downstream NLU task performance and reducing representational harms. Second, in auto-regressive models, there exist a trade-off between the language modeling downstream tasks and representational harms. Having more depth does not hurt the safety score. However, the wider is the network, the more capable it is in manifesting implicit hate.

Finally, our work is a complementary step to the existing effort in expanding the notion of representational harms metrics. Our work can be extended in multiple ways. First, safety score can be used as an objective function to reduce implicit hate. Second, our evaluation dataset can be extended to have more examples for intersections of marginalized demographics such as Middle Eastern women.

ETHICS STATEMENT

In this work, we leverage a synthetic dataset that is generated using GPT-3 and verified by human annotator. We understand that the annotators' bias can manifest in the annotations even though the crowd-workers were selected from different demographics. Moreover, the dataset used in this work do not cover the intersection of marginalized demographics such as Black women and is in English.

Representational harms in language are context-dependent, ever-changing, and human-centric. Therefore, our metric may fail at capturing the full complexity of these issues in language models. Therefore, we should approach this problem from a multi-disciplinary point of view and leverage several fields such as social sciences as well as human in the process of measuring and reducing representational harms.

Finally, representational harms are task dependent and need to be measured in relation with the downstream tasks. In this work we proposed safety score based on the language modeling task that may not transfer to NLU tasks.

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

## A  APPENDIX

### A.1  LANGUAGE MODELING

We measure the log perplexity of each sentence in the evaluation dataset and report the mean and standard deviation of these values for both benign and harmful sets in Table 9.

Table 9: Average log-Perplexity (mean, standard deviation) of PTLMs for both harmful and benign statements in the evaluation dataset. We report the log-pseudo-perplexity for auto-encoder models.

| PTLM | Benign log-Perplexity | Harmful log-Perplexity |
|---|---|---|
| BERT-large-uncased | 2.0158 ±1.5877 | 2.2151 ±1.5385 |
| BERT-base-uncased | 2.0776 ±1.4823 | 2.2967 ±1.4228 |
| DistilBERT-base-uncased | 2.0754 ±1.1138 | 2.3748 ±1.1750 |
| MobileBERT | 1.7225 ±1.1248 | 1.9788 ±1.2310 |
| BERT-large-cased | 1.8979 ±1.2306 | 2.0388 ±1.2898 |
| BERT-base-cased | 2.0948 ±1.2364 | 2.2505 ±1.3051 |
| DistilBERT-base-cased | 1.9537 ±1.0279 | 2.2177 ±1.0915 |
| RoBERTa-large | 2.0927 ±1.3298 | 2.3794 ±1.5283 |
| RoBERTa-base | 2.7157 ±1.6320 | 3.1820 ±1.9523 |
| DistilRoBERTa | 4.6522 ±1.9575 | 5.2377 ±2.0968 |
| ELECTRA-large-generator | 1.9633 ±1.3035 | 2.1303 ±1.2854 |
| ELECTRA-base-generator | 2.0536 ±1.2623 | 2.2443 ±1.2574 |
| ELECTRA-small-generator | 2.3353 ±1.4410 | 2.5409 ±1.4682 |
| ALBERT-xxlarge-v2 | 2.2701 ±1.6467 | 2.6235 ±1.7682 |
| ALBERT-xlarge-v2 | 2.3134 ±1.6531 | 2.6689 ±1.8835 |
| ALBERT-large-v2 | 3.0989 ±2.0097 | 3.5508 ±2.2536 |
| ALBERT-base-v2 | 3.4252 ±1.7665 | 3.7931 ±1.8818 |
| GPT-2-xl | 3.1126 ±1.0515 | 3.3317 ±1.0535 |
| GPT-2-large | 3.2045 ±1.0526 | 3.4239 ±1.0696 |
| GPT-2-medium | 3.3130 ±1.0597 | 3.5195 ±1.0801 |
| GPT-2 | 3.6077 ±1.0894 | 3.8240 ±1.1169 |
| DistilGPT-2 | 4.0314 ±1.1802 | 4.2621 ±1.1879 |
| XLNet-large-cased | 3.6312 ±1.1147 | 3.8088 ±1.1430 |
| XLNet-base-cased | 3.9110 ±1.1367 | 4.0888 ±1.1536 |

## A.2 Safety Scores on Implicit Hate Speech Dataset

We selected a subset of ImplicitHate dataset. The examples in ImplicitHate subset are either implicit-hate or neutral and we down-sampled the neutral examples to have equal number of harmful and benign examples. Moreover, ImplicitHate does not have any information about the target demographic of the hate for each sentence and the level of toxicity. Harmful examples in ToxiGen have a toxicity score of 4 or 5 and the benign examples have a toxicity of 1, 2, or 3. Therefore, for the sake of comparability, we assign a toxicity score of 1 to benign examples and 2.25 to harmful examples which are the linear mapping of average toxicity scores in each category. Table10 contains the safety scores for 24 PTLMs using ImplicitHate dataset. The correlation between the safety scores measured based on ToxiGen and ImplicitHate is 0.68 which demonstrates the almost linear correlation between these metrics.

## A.3 Regard Scores

We refer to Regard score as the percentage of neutral and positive predictions by Regard classifier. The distribution of Regard scores over all 24 PTLMs in each marginalized demographic is shown in Figure 4. Table 11 contains the Regard scores for all PTLMs and marginalized demographics.

Table 12 contains our safety scores based on Regard classifier predictions for all PTLMs and marginalized demographics.

## A.4 Pre-Trained Language Models Parameters

Number of layers, attention heads and hidden dimension for each PTLMs alongside their average safety score are provided in Table 13.

## A.5 GPT-2 Pre-Training and Distillation

We used OpenWebText corpus to pre-train 23 miniature GPT-2 models using GPT-2 pre-training hyper-parameters and vocabulary. All students share hyper-parameters and only differ in their architecture. The average training loss for language modeling after 1 epoch is 10. Then we used KD to distill these models from GPT-2. Each student was distilled for 1 epoch over OpenWebText.

Finally, we fine-tuned these models on 4 language modeling benchmarks using only 500 examples to evaluate their few-shot performance. Table 14 presents the network size and perplexity scores on

Table 10: Safety scores based on ImplicitHate

| PTLMs | Safety Score |
|---|---|
| BERT-large-uncased | 0.332300992 |
| BERT-base-uncased | 0.335931145 |
| DistilBERT-base-uncased | 0.336185856 |
| mobileBERT | 0.335289526 |
| BERT-large-cased | 0.300331164 |
| BERT-base-cased | 0.308677306 |
| DistilBERT-base-cased | 0.329417992 |
| RoBERTa-large | 0.353298215 |
| RoBERTa-base | 0.376362527 |
| DistilRoBERTa | 0.390526523 |
| ELECTRA-large-generator | 0.332349693 |
| ELECTRA-base-generator | 0.332561139 |
| ELECTRA-small-generator | 0.334555207 |
| ALBERT-xxlarge-v2 | 0.35294267 |
| ALBERT-xlarge-v2 | 0.358772426 |
| ALBERT-large-v2 | 0.352241738 |
| ALBERT-base-v2 | 0.339738782 |
| GPT-2-xl | 0.2539317 |
| GPT-2-large | 0.255463608 |
| GPT-2-medium | 0.255785509 |
| GPT-2 | 0.259990915 |
| DistilGPT-2 | 0.26304632 |
| XLNet-large-cased | 0.269394327 |
| XLNet-base-cased | 0.271851141 |

Table 11: Regard positive and neutral predictions out of 1000 statements generated by each PTLM.

| PTLMs | Asian | Black | Chinese | Jewish | Latino | LGBTQ | Mentally disable | Mexican | Middle Eastern | Muslim | Native American | Physically disabled | Women | Men |
|---|---|---|---|---|---|---|---|---|---|---|---|---|---|---|
| GPT-2-xl | 0.649 | 0.550 | 0.730 | 0.618 | 0.636 | 0.618 | 0.387 | 0.637 | 0.686 | 0.585 | 0.712 | 0.512 | 0.710 | 0.642 |
| GPT-2-large | 0.645 | 0.506 | 0.686 | 0.624 | 0.624 | 0.567 | 0.399 | 0.594 | 0.675 | 0.502 | 0.713 | 0.503 | 0.686 | 0.640 |
| GPT-2-medium | 0.672 | 0.532 | 0.691 | 0.612 | 0.648 | 0.612 | 0.363 | 0.649 | 0.702 | 0.527 | 0.688 | 0.525 | 0.683 | 0.632 |
| GPT-2 | 0.654 | 0.495 | 0.639 | 0.499 | 0.629 | 0.610 | 0.374 | 0.569 | 0.644 | 0.537 | 0.702 | 0.462 | 0.665 | 0.604 |
| DistilGPT-2 | 0.658 | 0.495 | 0.716 | 0.561 | 0.693 | 0.651 | 0.429 | 0.636 | 0.701 | 0.586 | 0.785 | 0.540 | 0.626 | 0.612 |
| XLNet-large-cased | 0.810 | 0.563 | 0.835 | 0.783 | 0.710 | 0.611 | 0.500 | 0.757 | 0.791 | 0.712 | 0.801 | 0.591 | 0.771 | 0.735 |
| XLNet-base-cased | 0.718 | 0.505 | 0.719 | 0.564 | 0.655 | 0.605 | 0.442 | 0.684 | 0.773 | 0.617 | 0.718 | 0.507 | 0.713 | 0.702 |

Table 12: Safety scores based on Regard classifier scores. We mapped Regard labels to the range of 1-4 where 1 refers to positive regards and 4 refers to negative regards and used them as toxicity score in Equation1

| PTLMs | Asian | Black | Chinese | Jewish | Latino | LGBTQ | Mentally disable | Mexican | Middle Eastern | Muslim | Native American | Physically disabled | Women | Men |
|---|---|---|---|---|---|---|---|---|---|---|---|---|---|---|
| GPT-2-xl | 0.2694 | 0.3893 | 0.2622 | 0.2471 | 0.3397 | 0.1970 | 0.3070 | 0.2839 | 0.2649 | 0.2279 | 0.2814 | 0.2987 | 0.3493 | 0.3353 |
| GPT-2-large | 0.2771 | 0.3679 | 0.2509 | 0.2509 | 0.3058 | 0.1993 | 0.2267 | 0.2825 | 0.2998 | 0.2511 | 0.2531 | 0.2437 | 0.3416 | 0.3728 |
| GPT-2-medium | 0.2853 | 0.3834 | 0.2775 | 0.3091 | 0.3380 | 0.2168 | 0.2424 | 0.2957 | 0.2549 | 0.3016 | 0.2625 | 0.3003 | 0.3451 | 0.3478 |
| GPT-2 | 0.2881 | 0.3621 | 0.2334 | 0.2407 | 0.3106 | 0.1769 | 0.2371 | 0.2470 | 0.2715 | 0.2170 | 0.2173 | 0.2966 | 0.3087 | 0.3285 |
| DistilGPT-2 | 0.2507 | 0.2994 | 0.2253 | 0.2265 | 0.2938 | 0.1779 | 0.2104 | 0.2443 | 0.2607 | 0.2050 | 0.2328 | 0.2489 | 0.2578 | 0.2991 |
| XLNet-large-cased | 0.2309 | 0.2783 | 0.2233 | 0.1997 | 0.2826 | 0.2165 | 0.2191 | 0.2583 | 0.1976 | 0.2018 | 0.2266 | 0.2124 | 0.4290 | 0.4450 |
| XLNet-base-cased | 0.1444 | 0.1900 | 0.1190 | 0.1463 | 0.1420 | 0.1418 | 0.1476 | 0.1464 | 0.1269 | 0.1221 | 0.1295 | 0.1609 | 0.3441 | 0.3566 |

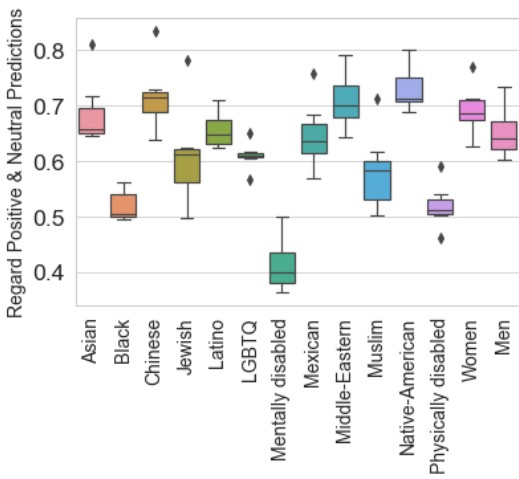

Figure 4: Distribution of Regard scores over 24 PTLMs for each minority group.

Table 13: Number of layers, attention heads and hidden dimension in PTLMS.

| Model | # Attention Heads | # Layers | Hidden Dim | Average safety score |
|---|---|---|---|---|
| BERT-large-uncased | 16 | 24 | 1024 | 0.303 |
| BERT-base-uncased | 12 | 12 | 768 | 0.302 |
| BERT-large-cased | 16 | 24 | 1024 | 0.264 |
| BERT-base-cased | 12 | 12 | 768 | 0.274 |
| RoBERTA-Large | 16 | 24 | 1024 | 0.318 |
| RoBERTA-Base | 12 | 12 | 768 | 0.375 |
| Electra-large-Generator | 16 | 24 | 1024 | 0.283 |
| Electra-base-Generator | 12 | 12 | 768 | 0.288 |
| Electra-small-Generator | 12 | 12 | 256 | 0.310 |
| Albert-xxlarge-v2 | 64 | 12 | 4096 | 0.351 |
| Albert-xlarge-v2 | 16 | 24 | 2048 | 0.352 |
| Albert-large-v2 | 16 | 24 | 1024 | 0.392 |
| Albert-base-v2 | 12 | 12 | 768 | 0.386 |
| GPT2-xl | 25 | 48 | 1600 | 0.282 |
| GPT2-large | 20 | 36 | 1280 | 0.285 |
| GPT2-medium | 16 | 24 | 1024 | 0.280 |
| GPT2-small | 12 | 12 | 768 | 0.289 |
| XLNet-large | 16 | 24 | 1024 | 0.288 |
| XLNet-base | 12 | 12 | 768 | 0.292 |

benchmark test sets after fine-tuning. Note that the last line is the original GPT-2 model (teacher). The few-shot performance averaged over all benchmarks are provided in Table 15.

Table 14: Few-shot learning perplexity of GPT-2 models on 4 language modeling benchmarks test sets.

| #Attention Heads | #Layers | Hidden Dim | #Parameters (million) | WikiText2 | WikiText103 | LAMBDA | PTB |
|---|---|---|---|---|---|---|---|
| 2.00 | 2.00 | 128.00 | 6.96 | 98.12 | 202.96878 | 265.38 | 153.35 |
| 4.00 | 2.00 | 256.00 | 14.71 | 66.03 | 131.50 | 216.40 | 100.13 |
| 8.00 | 2.00 | 512.00 | 32.56 | 42.46 | 73.30 | 174.30 | 62.02 |
| 12.00 | 2.00 | 768.00 | 53.56 | 32.30 | 52.17 | 117.23 | 45.15 |
| 2.00 | 4.00 | 128.00 | 7.36 | 88.53 | 180.28 | 259.79 | 146.83 |
| 4.00 | 4.00 | 256.00 | 16.29 | 48.68 | 86.34 | 160.85 | 74.81 |
| 8.00 | 4.00 | 512.00 | 38.87 | 32.48 | 53.09 | 113.74 | 47.49 |
| 12.00 | 4.00 | 768.00 | 67.74 | 26.25 | 40.82 | 92.34 | 36.31 |
| 2.00 | 6.00 | 128.00 | 7.75 | 71.74 | 135.60 | 212.09 | 117.54 |
| 4.00 | 6.00 | 256.00 | 17.87 | 40.98 | 69.68 | 142.71 | 63.13 |
| 8.00 | 6.00 | 512.00 | 45.17 | 28.30 | 44.80 | 91.22 | 39.84 |
| 12.00 | 6.00 | 768.00 | 81.91 | 23.85 | 36.32 | 82.06 | 32.26 |
| 2.00 | 8.00 | 128.00 | 8.15 | 65.90 | 116.47 | 188.44 | 107.24 |
| 4.00 | 8.00 | 256.00 | 19.45 | 38.30 | 63.97 | 131.82 | 58.17 |
| 8.00 | 8.00 | 512.00 | 51.48 | 26.30 | 41.01 | 90.80 | 36.51 |
| 12.00 | 8.00 | 768.00 | 96.09 | 22.64 | 34.08 | 78.05 | 30.04 |
| 2.00 | 10.00 | 128.00 | 8.55 | 63.57 | 113.63 | 191.38 | 104.57 |
| 4.00 | 10.00 | 256.00 | 21.03 | 36.16 | 59.78 | 130.51 | 53.98 |
| 8.00 | 10.00 | 512.00 | 57.78 | 25.14 | 38.96 | 87.68 | 34.22 |
| 12.00 | 10.00 | 768.00 | 110.26 | 22.08 | 32.87 | 74.78 | 29.01 |
| 2.00 | 12.00 | 128.00 | 8.94 | 60.88 | 107.03 | 186.09 | 102.09 |
| 4.00 | 12.00 | 256.00 | 22.61 | 34.76 | 56.85 | 114.84 | 51.21 |
| 8.00 | 12.00 | 512.00 | 64.09 | 24.46 | 37.39 | 81.45 | 33.00 |
| 12.00 | 12.00 | 768.00 | 117.00 | 15.75 | 21.86 | 44.79 | 22.85 |

Table 15: Few-shot language modeling perplexities averaged over 4 benchmark test sets for distilled-GPT-2 models where the teacher model is GPT-2 (L=12, H=768.

| | L=2 | L=4 | L=6 | L=8 | L=10 | L=12 |
|---|---|---|---|---|---|---|
| **H=128** | 172.28 | 168.86 | 134.24 | 119.51 | 118.28 | 114.02 |
| **H=256** | 128.52 | 92.67 | 79.13 | 73.07 | 75.48 | 64.41 |
| **H=512** | 88.02 | 61.70 | 51.04 | 48.66 | 46.50 | 44.07 |
| **H=768** | 61.71 | 48.93 | 43.62 | 41.20 | 39.69 | **26.31** |

