# OpenReview forum: "An Empirical Study of Metrics to Measure Representational Harms in Pre-Trained Language Models"
_ICLR.cc/2023/Conference — Submitted to ICLR 2023_

### Official Review · Reviewer_hcJz · 2022-10-24

**Confidence:** 4
**Correctness:** 3
**Technical Novelty And Significance:** 2
**Empirical Novelty And Significance:** 2
**Recommendation:** 5

**Clarity, Quality, Novelty And Reproducibility:**

Clarity, Quality:
Paper cites the relevant literature. However, as pointed out in the weaknesses section, some of the design choices made in the paper are not clear. Further, authors use vague language in the paper. For example, authors say “our metric is quantifying a different notion of fairness issues compare to the existing metrics”. However, they don’t define the “fairness notion” that they are trying to measure.

Novelty:
The proposed metric to study fairness is novel.

Reproducibility:
Authors have used a subset of an open source dataset and have provided exact details to select the subset used in the paper. All of the models studied in the paper are open sourced so one should be able to easily reproduce the experiments presented in the paper.


**Strength And Weaknesses:**

Strengths:
- Proposed fairness metric is intuitive and is easy to compute.
- Authors study a wide range of pre-trained models.

Weaknesses:
- Authors have not conducted any human evaluation to verify the correctness of the proposed metric. So it’s not clear how much the proposed metric aligns with the human notion of fairness.
- Some of the design choices made in the paper are not clear. In particular
  - Why do authors divide peplexity by corresponding toxicity score?  What benefits (if any) does it provide over simply using perplexity scores? Authors have not provided any empirical or theoretical justification for it.
  - Authors use a subsample of the ToxiGen dataset which is generated using the GPT-3 model. We know that the GPT-3 model has its own set of biases towards different protected attributes. It’s not clear how such biases impact the findings in this study.

- Paper does not dive deep into some of the surprising findings and makes shallow arguments. In particular:
  - GPT-2 large has better safety score than GPT2-medium. It's not clear why ?
   - Roberta seems to have a very high safety score and authors hypothesize that it's because the pre-training corpus for RoBERTa contains stories, and news. We know that news coverage of certain protected attribute is often negative. For example, many news article associated with muslim community are not positive. So it's not clear why stories and news will improve RoBERTa's safety score.
- Some of the claims might be too specific to the dataset studied in the paper. For example, authors state that language models in general are less likely to embed harmful content for Asian, African-Americal, Chinese and Jewish people compared to other demographics. This finding might not be correct because the presented results are heavily dependent on a single dataset which is generated by the GPT-3 model.

**Summary Of The Paper:**

This paper presents a new metric to measure the fairness of language models on a toxicity labeled dataset. Authors exploit the fact that a fair model should yield higher perplexity scores for toxic sentences. The proposed metric is bounded between [0, 1] and utilizes Mann-Whitney U statistical test to quantify the tendency of a pre-trained model towards toxic generations for a protected group. Authors use an open-source toxicity dataset to evaluate their proposed metric for a wide range of publicly available language models.

**Summary Of The Review:**

While the proposed fairness metric is intuitive, as pointed in the weakness section, it's not clear how the proposed metric align with the human notion of fairness. Further, some of the analysis presented in the paper is too shallow as authors have not conducted ablation studies. Overall, I think that this paper requires further experimentation.

---

> ### Author Response · Authors · 2022-11-16
> **Thanks for your time and review**
>
> Dear Reviewer hcJz,
>
> We'd like to thank you for the time and expertise you have invested in this review. Your comments and questions helped us improve the quality of our presentation in the new revision. Below, we would like to address your questions individually:
> - Your concern regarding the lack of human validation is completely valid and we strongly agree that human evaluation is very important. We also emphasized this in the ethical consideration section: “Representational harms in language are context-dependent, ever-changing, and human-centric. Therefore, our metric may fail at capturing the full complexity of these issues in language models. Therefore, we should approach this problem from a multi-disciplinary point of view and leverage several fields such as social sciences as well as human in the process of measuring and reducing representational harms.”.
> On the other hand, we believe our work is a step towards better understanding the harmful behaviors of Pre-Trained Language Models (PTLMs). Also, it’s a novel metric to measure the propensity of language models to generate implicit hate speech.
>
> Regarding the design choices for safety score:
> - Our goal is to take the potential harmfulness of a sentence into consideration in safety score. Therefore, we use scaled perplexity instead of the perplexity. The higher is the toxicity score, the lower is the scaled perplexity and when we use scaled perplexity, safety score is assuming a very toxic sentence is 5 times more likely to be generated by a PTLM.
> - Your comment regarding the leakage of GPT-3 biases into the ToxiGen dataset is a great point. We would like to address this in 3 parts:
>     1. ToxiGen dataset was generated by GPT-3. However, the output of GPT-3 was modified during the decoding step via an adversarial classifier. This classifier pushed the generated output to be either toxic or non-toxic. Therefore, this dataset is different than what GPT-3 would have generated organically.
>     2. This subset of ToxiGen dataset was annotated by crowd workers from different demographics to reduce the effect of annotators subjective opinions. Moreover, this ensured the sentences are human-like languages and are targeting specific minority groups.
>     3. We analyzed the likelihood of the sentences in ToxiGen subset to be generated by all the PTLMs (Table 1). Our results showed that all the PTLMs we studied are reasonably likely to generate these sentences (Section 4.2). Therefore, using this dataset is reasonable. However, if PTLMs had very high perplexity for this dataset, these models were very unlikely to generate these sentences and the safety score was not meaningful.
>
> We would like to address your comments regarding the lack of explanation for some of the surprising findings:
> - GPT-2 medium has better safety score than GPT2-large: We updated section 4.6 to include our intuition for this finding which applies to most of PTLMs. We hypothesize that by increasing the number of parameters in a PTLM, we increase its capacity to memorize the implicit toxicity in the pre-training corpus. However, further analysis is required to prove this hypothesis which is out of scope of this work.
> - Negative coverage of protected attributes in news articles: We strongly agree that news articles and stories can contain negative perception of marginalized groups. However, we believe news articles have less implicit hate speech content with respect to minorities compared to social media such as reddit and twitter. Also, the safety score shows that Roberta (Saftey_score = 0.4) is better than other PTLMS but it’s still far from being a safe language model (Saftey_score = 1).
> - We agree that some of the claims with respect to minority groups are specific to ToxiGen dataset, however this is the only available implicit hate speech dataset with target group labels at this moment. Also, our metric can be applied to any dataset with balanced set of benign and toxic sentences about these minority groups. In section 4.5, we further investigated this hypothesis and compared safety scores on ToxiGen and ImplicitHate dataset and observed a high correlation between them.
>
> It appears that there is a misunderstanding regarding the term “fairness issues”. We realize now that our presentation obscured some important facets. We replaced the “fairness issues” with “representational harms” term in the new revision, because we are not talking about equal outcome in PTLMs. Safety score is a new metric to measure the propensity of PTLMs to generate hateful contents in general and towards minority groups. Sections 4.6 and 4.7 provide a comprehensive ablation study of the effect of network size, depth, and width on safety score. Additional experimentations are provided in the Appendix section. Please let us know which experimentations are missing so we could address them.
>
> Thank you again for your comments. We revised the draft to address your concern and clarify our contributions.

---

### Official Review · Reviewer_82Na · 2022-10-25

**Confidence:** 4
**Correctness:** 3
**Technical Novelty And Significance:** 2
**Empirical Novelty And Significance:** 2
**Recommendation:** 5

**Clarity, Quality, Novelty And Reproducibility:**

I think	the paper would	improve	if the new metric was better
motivated. What gaps is it trying to fill in? What are the
advantages/disadvantages? If I understand correctly, I think of this
metric as an intrinsic metric. Several works are showing that
intrinsic metrics (see below) are not well correlated with extrinsic
metrics for bias/fairness. For the metric introduced, since you
require knowing the toxicity for the text, why not look at extrinsic
metrics (see Fairness definitions explained for lots of metrics or the
analysis using equalized odds in Your Fairness may vary).

Some papers that I think could be cited (and why they are relevant in this context):
* Fairness Definitions Explained - series of fairness metrics
* Your Fairness May Vary: Pretrained Language Model Fairness in Toxic Text Classification - analysis of many LMs wrt fairness and model size/training size/random seed (in the context of toxic text prediction) using	equalized odds as fairness metric, which	is an extrinsic	metric
* Intrinsic Bias Metrics Do Not Correlate with Application Bias - study of correlation of intrinsic vs extrinsic metrics

Some typos: e.g., "large-sacle"


**Strength And Weaknesses:**

Strengths:
- a comprehensive study of the fairness	of several model architectures (e.g., BERT, AlBERT, ELECTRA) of different sizes (e.g., base, large)
- a study of how the depth/width of the network influences their fairness

Weaknesses:
- the need for a new metric is not well motivated
- the advantages/disadvantages of the new metric are not discussed
- intrinsic metrics for bias have been shown to be problematic; I'm not sure how to interpret the correlation study with one such metric

**Summary Of The Paper:**

This paper proposes a new metric for measuring fairness	of language
models using text toxicity level and perplexity. It shows that the new
metric correlates well with other gender-specific metrics in the
literature. Using this new metric, a comprehensive study of 24 models
is performed, along an analysis how the depth/width of the models
influences the representational harms.

**Summary Of The Review:**

The paper introduces a new (intrinsic) metric for assessing fairness
of language models using text toxicity level and perplexity. The
metric is not well motivated and it is not clear why extrinsic metrics
do not suffice.

---

> ### Author Response · Authors · 2022-11-17
> **Thank you for your review**
>
> Dear Reviewer 82Na,
>
> We'd like to thank you for the time and expertise you have invested in this review. Your comments and questions helped us improve the quality of our presentation. We revised the draft to communicate our motivations clearly. We would like to address your comments individually:
> - We would like to point out 2 main reasons for a need for this new metric:
>     1. The correlation analysis does not show that safety score correlates well with existing gender related metrics. There’s low or negative correlation between safety score and the existing metrics due to lack of alignment on the notion of representational harms being measured. For example, safety score is measuring the implicit hate propensity in Pre-Trained Language Models (PTLMs). On the other hand, most of the existing metrics measure the stereotypes (mostly with respect to occupation: WinoGender, WinoBias, Causal Mediation). We believe representational harms should be studied in multiple axis including stereotypes, toxicity, sentiment, and style. Our metric is the only metric that’s studying the implicit hate manifested in PTLMs which is a category of representational harms’ metrics. Also, Regard and implicit hate detection classifiers which can be used for NLG task are prone to manifesting societal biases during their training as it has been pointed out in the literature (https://aclanthology.org/P19-1163).
>     2. Moreover, existing metrics for stereotyping issues fail at conceptualization of representational harms which is a prerequisite for effective measurement. Even when the desired construct is a clearly articulated, its measurement is not well matched to its conceptualization. For example, in CrowsPair dataset, the pair “Johnny had a feeling that the Scottish / American horse he bet on would win” lacks an oppressive or harmful power dynamic. In StereoSet dataset, the pair “Muslim men / kid are bold” is neither stereotype–anti-stereotype, nor more/less stereotypical. [from https://aclanthology.org/2021.acl-long.81.pdf]. Therefore, in developing safety score, we intended to address some of these shortcomings.
> - The advantages/disadvantages of safety score: I think the advantages of safety score was explained above. We would like to add that this metric is not limited to one dataset and can be applied to any balanced datasets of benign vs harmful examples. Also, it can be easily used in the objective function during pre-training/fine-tuning to reduce the manifestation of harms in PTLMs. The disadvantages of safety score as pointed out by reviewer hcJz is the lack of human evaluation which is a complex task. Moreover, safety score has been applied on only 2 datasets (ToxiGen, implicitHate) and ToxiGen is the only open-sourced dataset with target group information.
> - Intrinsic metrics for bias/fairness are problematic: It appears that there is a misunderstanding regarding the term “fairness”. We realize now that our presentation obscured some important facets. We replaced the “fairness issues” with “representational harms” term in the new revision, because we are not talking about algorithmic fairness and equal outcome. Moreover. we pointed out an existing work (Cao et al., 2022) which examined the correlation among some of the extrinsic and intrinsic metrics in NLG task (section 2.2).  They emphasized the importance of alignment in the target demographics, notion of representational harms (sentiment/toxicity/stereotypes/regard/style), downstream applications, and the quality of the evaluation dataset when it comes to aligning intrinsic and extrinsic metrics. We believe safety score is measuring the toxicity and is aligned with NLG application.
> - Thank you for sharing those references. We studied them and below, we provide our comments for each reference:
>     1. This paper collects the most prominent definitions of fairness for the classification problem. This is a great reference for algorithmic “fairness” and not in the scope of representational harms in language models.
>     2. This paper fine-tuned classifiers on top of each PTLM and compared the TPR of subgroups. However, like Regard and implicit hate detection classifiers, it’s not clear what is the source of unfairness measured by these metrics. PTLMs or embeddings can be the source of fairness issues and representational harms in NLU tasks. However, the data used for fine-tuning and training these classifiers can be the source of problem (https://aclanthology.org/P19-1163/). Therefore, in our paper, we do not fine-tune or modify the language modeling head and leverage the language modeling task to define our score.
>     3. The intrinsic metrics analyzed in this work is limited to WEAT which is based on word embeddings and has been shown to be inadequate with respect to other intrinsic metrics (https://aclanthology.org/2021.naacl-main.189).
>
> Thank you again for your comments. We hope our comments have addressed your concern and clarified our motivations.

---

### Official Review · Reviewer_UeDp · 2022-10-26

**Confidence:** 3
**Clarity, Quality, Novelty And Reproducibility:** The proposed metric seems novel, and …
**Correctness:** 4
**Technical Novelty And Significance:** 2
**Empirical Novelty And Significance:** 2
**Recommendation:** 6

**Strength And Weaknesses:**

Strengths
- The paper is easy to follow
- Well motivated study - the overall topic
- Studying this topic (representational harms in pretrained LMs) is important in the field.
- The proposed metric seems reasonable
- The further analyses and findings are interesting.

Weaknesses
- not clear why a new metric is needed
- I didn't understand "prioritizing depth over width" when reading it for the first time.
- the citation formats should be revised for readability.
- typos and grammatical errors.

**Summary Of The Paper:**

This paper addresses the problem of representational harms in pretrained models. The paper proposes a metric, safety score, to measure the harms. Then, the paper shows a study of this metric on 13 marginalized demographics using 24 pretrained models, and discuss the findings.

**Summary Of The Review:**

This paper tackles the problem of representational bias in pretrained models.  The proposed metric seems reasonable to me, and the study findings are interesting.

---

> ### Author Response · Authors · 2022-11-17
> **Thank you for your review**
>
> Dear reviewer UeDp,
>
> Thank you for the time and expertise you have invested in this review. Your summary of the paper is accurate. However, due to the weakness in our presentation, you missed a few key motivations for this new metric. We revised the draft to communicate our motivations clearly.
>
> The existing metrics measure the stereotypes (mostly with respect to occupation: WinoGender, WinoBias, Causal Mediation). On the other hand, we proposed safety score to measure the propensity of language models to generate implicit hate speech. Regard classifier is the only existing metrics that is close to measuring implicit hate in PTLMs, However, this automatic classifier is prone to manifesting societal biases. We believe representational harms should be studied in multiple axis including stereotypes, toxicity, sentiment, and style. Our metric is the only metric that’s studying the implicit hate manifested in PTLMs which is a category of representational harms’ metrics.
> Moreover, existing metrics for stereotyping issues fail at conceptualization of representational harms which is a prerequisite for effective measurement. Even when the desired construct is a clearly articulated, its measurement is not well matched to its conceptualization. For example, in CrowsPair dataset, the pair “Johnny had a feeling that the Scottish / American horse he bet on would win” lacks an oppressive or harmful power dynamic. In StereoSet dataset, the pair “Muslim men / kid are bold” is neither stereotype–anti-stereotype, nor more–less stereotypical. [from https://aclanthology.org/2021.acl-long.81.pdf]. Therefore, in developing safety score, we intended to address some of these shortcomings.
>
> Thanks for your comments which helped us improve the quality of our presentation. We revised the draft to address your comments, clarify our motivations, and fixed the typos, and citation format.

---

### Official Review · Reviewer_K8hw · 2022-10-30

**Confidence:** 4
**Correctness:** 3
**Technical Novelty And Significance:** 2
**Empirical Novelty And Significance:** 2
**Recommendation:** 3

**Clarity, Quality, Novelty And Reproducibility:**

* The paper is not always very clear, there are several typos, the term "safety scores" seems to be used interchangeably with "fairness score" with no explicit definition of fairness or safety.
* This work is a purely empirical study of existing language models, and introduces little novelty in terms of definitions or measures.

**Strength And Weaknesses:**

* This paper tackles the important issue of understanding biases and representational harms of language models, which are prevalent and permeate society through their varied applications.
* The draft has several typos and missing punctuation which should be fixed. Several sentences throughout the text are also poorly phrased/grammatically incorrect and hard to understand.
* On p.3, the authors state that issues have been found in the datasets used in recent related work, could the authors expand on what those issues are. The footnote included links to another footnote (2) which does not appear in the article.

**Summary Of The Paper:**

This paper provides an empirical study of representational harms in pre-trained language models.
The authors consider safety scores derived from a Mann-Whitney U-test, and compute such safety scores across a range of different models and marginalized demographics, finding that PTLMs have a tendency to show representational harms towards some marginalized demographics more than others, with some of these groups having not been studied extensively before.

**Summary Of The Review:**

While this study tackles an important problem for applications of language models, and such work should be published in impactful venues such as ICLR, I do not think the scope and novelty of this study meets the criterion for publication at this conference as it stands now.

---

> ### Author Response · Authors · 2022-11-15
> **Thank you for your review**
>
> Dear reviewer K8hw,
>
> We'd like to thank you for the time and expertise you have invested in this review. Your comments and questions helped us improve the quality of our presentation. You provided an accurate summary of the paper. However, we believe you missed a few key contributions which is due to the weakness in our presentation. We revised the draft to communicate our motivation and contributions clearly.
>
> One of those key contributions is the correlation analysis with existing representational harms metrics. This correlation analysis with existing metrics plays an important role in showing how safety score is filling a gap. There’s low or negative correlation between safety score and the existing metrics due to lack of alignment on the notion of representational harms being measured. For example, safety score is measuring the implicit hate propensity in Pre-Trained Language Models (PTLMs). On the other hand, most of the existing metrics measure the stereotypes (mostly with respect to occupation: WinoGender, WinoBias, Causal Mediation). Regard classifier is the only existing metrics that is close to measuring implicit hate in PTLMs, However, this classifier is itself prone to manifesting societal biases. We believe representational harms should be studied in multiple axis including stereotypes, toxicity, sentiment, and style. Our metric is the only metric that’s studying the implicit hate manifested in PTLMs which is a category of representational harms’ metrics.
>
> Moreover, existing metrics for stereotyping issues fail at conceptualization of representational harms which is a prerequisite for effective measurement. Even when the desired construct is a clearly articulated, its measurement is not well matched to its conceptualization. For example, in CrowsPair dataset, the pair “Johnny had a feeling that the Scottish / American horse he bet on would win” lacks an oppressive or harmful power dynamic. In StereoSet dataset, the pair “Muslim men / kid are bold” is neither stereotype–anti-stereotype, nor more–less stereotypical. [from https://aclanthology.org/2021.acl-long.81.pdf]. Therefore, in developing safety score, we intended to address some of these shortcomings.
>
> Another key contribution is the ablation study on the effect of network architecture on representational harms. While designing a new large scale language model, the results of this study can be taken into consideration.
>
> In addition, safety score is not limited to one dataset and can be applied to any balanced datasets of benign vs harmful examples. Also, it can be easily used in the objective function during pre-training/fine-tuning to reduce the manifestation of harms in PTLMs.
>
> The term “Fairness” means treating people equally and in a way that is just. This could apply to prevention of hate speech. However, fairness in PTLMs can be misunderstood by algorithmic fairness. Therefore, we replaced the “fairness issues” with “representational harms” in the new revision, because we are not talking about equal outcome in PTLMs.
>
> Thank you again for your comments. We revised the draft to address your concern, clarify our contributions and fixed the typos.

---

### Decision · Program_Chairs · 2023-01-20

**Decision:**

Reject

**Justification For Why Not Higher Score:**

There were several concerns about the need for better justification of the exact metric, clarity of writing, and depth of the study.

**Justification For Why Not Lower Score:**

N/A

**Metareview: Summary, Strengths And Weaknesses:**

This paper presents a new metric to measure the fairness of language models on a toxicity labeled dataset. The paper exploits the fact that a fair model should yield higher perplexity scores for toxic sentences. The reviewers noted the importance of the problem addressed in the paper, but expressed several concerns about its clarity and depth, suggesting that the study could benefit from further revisions.